# Automated Deep Learning for load forecasting

**Julie Keisler**[1,2]  **Sandra Claudel**[1]  **Gilles Cabriel**[1]  **Margaux Brégère**[1,3]

[1]EDF R&D, Lab Paris-Saclay
[2]INRIA Lille Nord Europe
[3]LPSM, Sorbonne Université

**Abstract**  Accurate forecasting of electricity consumption is essential to ensure the performance and stability of the grid, especially as the use of renewable energy increases. Forecasting electricity is challenging because it depends on many external factors, such as weather and calendar variables. While regression-based models are currently effective, the emergence of new explanatory variables and the need to refine the temporality of the signals to be forecasted is encouraging the exploration of novel methodologies, in particular deep learning models. However, Deep Neural Networks (DNNs) struggle with this task due to the lack of data points and the different types of explanatory variables (e.g. integer, float, or categorical). In this paper, we explain why and how we used Automated Deep Learning (AutoDL) to find performing DNNs for load forecasting. We ended up creating an AutoDL framework called EnergyDragon[1] by extending the DRAGON[2] package and applying it to load forecasting. EnergyDragon automatically selects the features embedded in the DNN training in an innovative way and optimizes the architecture and the hyperparameters of the networks. We demonstrate on the French load signal that EnergyDragon can find original DNNs that outperform state-of-the-art load forecasting methods as well as other AutoDL approaches.

## 1 Introduction

Currently, large-scale electricity storage is expensive and relies on inefficient systems. To ensure the safety and smooth operation of the electricity system, it is critical to maintain a strict balance between production and load at all times. Managing this balance relies primarily on the flexibility of programmable power plants which can anticipate electricity demand and adjust their activity accordingly. Load forecasting is essential to program these power plants and to ensure grid stability. Every year, power system operators need forecasting models to provide them with load trends for the coming year and to serve as the basis for short-term forecasts. These models are based on various explanatory variables such as weather (temperature in particular has a strong impact on load) or calendar variables (e.g. load tends to vary between weekdays and weekends). Historical load can be used as a target to train these models for earlier periods, but the one-year forecast horizon makes it unusable as a model input. For this reason, statistical and machine learning methods typically used in time series forecasting are not efficient for this problem. Regression methods, on the other hand, work very well. Over the year, these initial models are then "re-calibrated" with adaptive online learning methods (e.g., online expert aggregation, see Gaillard (2015) or Kalman filter, see Vilmarest (2022)) using the lagged data as it becomes available. For example, the re-calibration can be used for day-ahead forecasting to help scheduling production resources for the next day. The re-calibration part is beyond the scope of this paper, which focuses on the stationary model.

The models used in industry and winning load forecasting competitions (see Farrokhabadi et al. (2022) for a recent one) are regression-based models such as Generalized Additive Models (GAMs) or tree-based models. However, to improve performance and robustness, and to respond to new

---

[1]Our code is available here: `https://github.com/JulieKeisler/automl.git`
[2]`https://dragon-tutorial.readthedocs.io/en/latest/`

industrial challenges such as the integration of new data or the need to forecast at increasingly finer time steps, interest is growing in deep neural networks (DNNs). This is a natural step, as DNNs have proven to be highly effective in fields such as computer vision and natural language processing (NLP). The literature on load forecasting with DNNs mainly approaches it from a time series point of view, using recurrent networks on recently lagged load, which is not applicable in our case. Moreover, DNNs are known to be poorly efficient on tabular regression (Grinsztajn et al., 2022). In our case, the lack of available data (compared to computer vision or NLP datasets, for example) is an additional challenge. The variables used as inputs to the models also have a major impact on performance and may be different from those that work well for the regression models. Nevertheless, we were able to create a DNN with a specific set of explanatory variables that achieves good performance while being slightly below the state of the art. We turned to Automated Deep Learning (AutoDL) to improve on this first model.

In this paper, we explain how we were able to effectively use AutoDL for load forecasting. We tested several existing methods in the literature, which could not compete with the state-of-the-art, and finally developed our own AutoDL framework: EnergyDragon. It uses the search space of the DRAGON package (Keisler et al., 2024) (for DiRected Acyclic Graph Optimization), but includes some innovations such as an original feature selection efficient for load forecasting and a faster search algorithm. Our framework makes it possible to find DNNs that outperform the state of the art in load forecasting by optimizing both their architectures and hyperparameters. We demonstrate its performance on an industrial use case: French load. Finally, we designed EnergyDragon to be understandable and appealing to load forecasting experts who may be new to deep learning. In summary, our contributions are as follows:

- An explanation of our strategy for applying AutoDL to a real-world application, namely load forecasting.

- The AutoDL framework EnergyDragon, an extension of DRAGON (Keisler et al., 2024) for load forecasting applications.

- A new feature selection method, embedded in the training of DNNs, that is efficient for load forecasting.

- An application of our results to a concrete use case: the French load forecasting. We show that our approach outperforms the state-of-the-art and other AutoDL techniques.

We begin this paper by presenting in Section 2 why and how we applied AutoDL to load forecasting and position ourselves with the literature. In Section 3, we introduce the design of EnergyDragon, an AutoDL framework for load forecasting. Finally, Section 4 details our experimental results obtained on a real-world use case: the forecast of the French load. Section 5 concludes the paper and presents further research opportunities.

## 2 Deep Learning and AutoDL for load forecasting

The load signal can be explained almost entirely by a set of explanatory variables that do not include past data. Therefore performing models tend to be based on regression rather than time series techniques. Multiple linear regressions (MLRs) can be used to calculate the relationships between multiple variables. However, the relationships between load and some exogenous variables are not linear and these models require the specification of functional forms for these variables. The generalized additive models (GAMs) for example, model the nonlinear effects using a spline basis (Pierrot and Goude, 2011). These models, highly accurate for load forecasting, are used in industry and have won several competitions (see for example Nedellec et al. (2014)). In this paper, we are interested in DNNs for load forecasting. Many existing works are based on a setting where past load

is immediately available and use time series techniques. For example, Sehovac and Grolinger (2020) uses a sequence-to-sequence recurrent network on historical load with data every five minutes, Rahman et al. (2018) and Mamun et al. (2019) use LSTM (Long-Short Term Memory) models on lagged data and temperature for day-ahead forecasting. Novaes et al. (2021), Zhou et al. (2021a) and L'Heureux et al. (2022) tried a transformer-based load forecaster using historical and calendar data for residential load data. Other works, closer to our setting, use DNNs with more explanatory variables or for longer forecast horizons. For example, Farsi et al. (2021) and He (2017) use parallel LSTM/CNN (Convolutional Neural Network) models with different forecast horizons and features, and del Real et al. (2020) forecasts the French load using temperature grids and calendar features as inputs within a CNN. Among all the proposed models, we built a competitive DNN based on CNN and MLP (Multi-Layer Perceptron) layers, called CNN/MLP in the following, whose architecture is closed to Farsi et al. (2021) and He (2017) (we detail this architecture in Section A.1).

Finding better DNNs for a given task can be done with Automated Deep Learning. AutoDL is a branch of Automated Machine Learning (AutoML) whose goal is to automatically find the best possible DNN for a given problem. AutoDL itself consists of two subproblems, the search for the best architecture, called Neural Architecture Search (NAS), and, for a fixed architecture, the search for the best hyperparameters, called HyperParameters Optimization (HPO). NAS approaches require the definition of a good search space representing all possible solutions. Most search spaces from the literature (Hutter et al., 2019) offer to optimize architectures suitable for computer vision tasks based on CNN layers, pooling layers, and skip connections. Closer to our setting, the AutoPytorch framework has been introduced for tabular (Zimmer et al., 2021) and time series (Deng et al., 2022) data. We tested this framework on our problem (see Section 4), which could not beat the CNN/MLP model. Then, inspired by Chen et al. (2021) and their work on NAS for multivariate time series forecasting, we used the DARTS (for Differential-Architecture Search, see the original paper Liu et al. (2018b)) to relax our original architecture (our search space is given Section B). Encouraged by the good results of DARTS, we further relaxed our search space using the DRAGON framework proposed by Keisler et al. (2024), originally introduced for time series forecasting. Compared to the DARTS approach, where the number of layers and the hyperparameters are fixed, the search space defined by DRAGON is more flexible (see Section 3.1).

Based on this framework, we created EnergyDragon, which uses the search space of DRAGON, but includes candidate operations specifically designed for load forecasting (see Section A.2) as well as an innovative feature selection method. The CNN/MLP based architecture depends highly on the input variables. Most AutoDL approaches do not address this issue, which is irrelevant in computer vision or NLP. Surprisingly, neither does Auto-Pytorch, while Grinsztajn et al. (2022) identified the lack of robustness of models to non-informative features as one of the reasons why DNNs perform poorly on tabular data compared to tree-based models. Outside the AutoDL community, feature selection is a widely discussed topic in the literature. Typical approaches include filter methods, wrapper methods, and embedded methods (Li et al., 2017). Filter methods select features based on statistical measures. Wrapper methods train the models with multiple subsets of features and evaluate the features importance based on performance. They are more computationally expensive than filter methods, but can be more efficient. Finally, embedded methods integrate feature selection into the model training process by penalizing the contribution of less important features. In this work, we took inspiration from the DARTS framework and developed our own embedded method, which is described in detail in Section 3.2.

## 3 EnergyDragon

In this section, we describe EnergyDragon, our DNNs optimization framework for load forecasting. Section 3.1 briefly presents the search space used, which is that of Keisler et al. (2024). Next, the following subsections details our contributions to the original framework, adapting it to load forecasting. In Subsection 3.2, we present the objective function. It covers not only network

evaluation, but also the feature selection. Due to the specific setting used for the load forecasting task, different from the time series used in Keisler et al. (2024), we had to restrained the search space defined Subsection 3.1 using a meta-architecture presented Subsection 3.3. Finally, Subsection 3.4 introduces our search algorithm, an asynchronous evolutionary algorithm.

## 3.1 Search Space

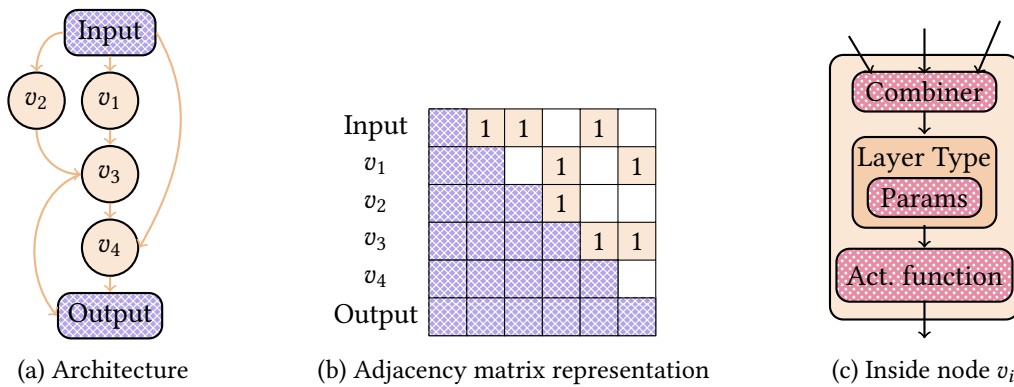

(a) Architecture  (b) Adjacency matrix representation  (c) Inside node $v_i$

Figure 1: DNN encoding as a directed acyclic graph (DAG), as proposed by Keisler et al. (2024).

The search space used in our framework was proposed by Keisler et al. (2024) and is defined as $\Omega = (\mathcal{A} \times \{\Lambda(\alpha), \alpha \in \mathcal{A}\})$, where $\mathcal{A}$ is the set of all considered architectures and $\Lambda(\alpha)$ is the set of all considered hyperparameters induced by the architecture $\alpha$. Each architecture $\alpha \in \mathcal{A}$ is represented by a DAG $\Gamma$, where the nodes are the DNN layers and the edges are the connections between them (see Figure 1a). The graph adjacency matrix $\mathcal{M} \in \mathbb{R}^{m \times m}$ is used to encode $\Gamma$, where $m$ is the number of nodes (see Figure 1b), along with a sorted list containing the node hyperparameters $\mathcal{L}$, where $|\mathcal{L}| = m$. In summary, $\mathcal{A} = \{\Gamma = (\mathcal{M}, \mathcal{L})\}$. Each architecture $\alpha \in \mathcal{A}$ induces a hyperparameter search space $\Lambda(\alpha)$. The chosen hyperparameters of all layers from an architecture $\alpha$ are placed in a vector denoted as $\lambda \in \Lambda(\alpha)$. As shown in Figure 1c, the layer type: convolution, recurrence, identity, etc., belongs to the architecture search space $\mathcal{A}$, but the layer type parameters: filter size, output shape, etc., the combiner, and the activation function are part of $\Lambda(\alpha)$. The combiner is a function used to combine the multiple inputs of the node. The architecture search space allows multiple input connections, and the incoming vectors can have different shapes. They are combined by the combiner. See Keisler et al. (2024) for more information about this search space.

## 3.2 Objective function

Our objective is to find the DNN $\hat{f} \in \Omega$ having the lowest forecast error on a given load signal. We consider a load dataset $\mathcal{D}$, containing the load signal and the explanatory variables. For any subset $\mathcal{D}_0 = (X_0, Y_0)$, the forecast error $\ell_{MSE}$ is defined as:

$$\ell_{MSE} \colon \Omega \times \mathcal{D} \to \mathbb{R}$$
$$f \times \mathcal{D}_0 \mapsto \ell_{MSE}(f(\mathcal{D}_0)) = \ell_{MSE}(Y_0, f(X_0)) = \mathrm{MSE}(Y_0, f(X_0)).$$

Where MSE is the Mean Squared Error. Each DNN $f \in \Omega$ is parameterized by:

- $\alpha \in \Lambda$, its architecture, optimized by the framework.

- $\lambda \in \Lambda(\alpha)$, its hyperparameters, optimized by the framework, where $\Lambda(\alpha)$ is induced by $\alpha$.

- $\theta \in \Theta(\alpha, \lambda)$, the DNN weights, where $\Theta(\alpha, \lambda)$ is generated by $\alpha$ and $\lambda$ and optimized by gradient descent when training the model.

In the following, $T$ represents the number of days in the data set, $H$ the number of time steps within a day, and $F$ the number of available explanatory variables. The data set $\mathcal{D} = (X, Y)$ consists of $Y = \{\mathbf{y}_t\}_{t=1}^T \in \mathbb{R}^{T \times H}$ the target variable and $X = \{\mathbf{x}_t\}_{t=1}^T = \{\mathbf{x}_i\}_{i=1}^F \in \mathbb{R}^{T \times H \times F}$ the explanatory variables. The optimization aims to find an optimal subset of explanatory variables: $\hat{X} = \{\mathbf{x}_j\}_{j \in \mathcal{P}(\{1,\ldots,F\})} \subseteq X$. To do this, we introduce $\forall j \in [\![1, F]\!] : p_j \in \{0, 1\}$ such that

$$\mathbf{x}_j \in \hat{X} \Leftrightarrow p_j = 1 \,.$$

To use gradient descent to find the optimal features, our indicators $p = (p_1, \ldots, p_F)$ are relaxed:

$$w = \{\text{sigmoid}(w_j)\}_{j=1}^F \in [0, 1]^F \quad \text{with} \quad w_j \in \mathbb{R} \quad \text{and} \quad p_j = \mathbb{1}_{w_j > 0} \,.$$

We partition our time indexes into three groups of successive time steps and split accordingly $\mathcal{D}$ into three datasets: $\mathcal{D}_{train}$, $\mathcal{D}_{valid}$, and $\mathcal{D}_{test}$. After choosing an architecture $\alpha$ and a set of hyperparameters $\lambda$, the DNN $f^{\alpha,\lambda}$ is built and trained on $\mathcal{D}_{train}$. The training of $f^{\alpha,\lambda}$ is divided into two parts. We consider $E = E_w + E_\theta$ as the total number of training epochs. The number of epochs when the feature vector $w$ and the weights $\theta$ are jointly optimized is called $E_w$. Starting at epoch $E_w + 1$, $w$ is transformed to $p$ using the equation $p_j = \mathbb{1}_{w_j > 0}$. Then $\theta$ is optimized until the end of the training. Two different losses are used during the training. In the first part, when the current epoch $e \le E_w$, an L1 penalty is added to $\ell_{MSE}$, like in the LASSO regression (Tibshirani, 1996), to restrain the number of selected features. We define the joint model and features loss $\tilde{\ell}$ as:

$$\tilde{\ell} \colon \Omega \times [0, 1]^F \times \mathcal{D} \to \mathbb{R}$$

$$f_\theta^{\alpha,\lambda} \times w \times \mathcal{D}_0 \mapsto \ell_{MSE}\big(f_\theta^{\alpha,\lambda}(\mathcal{D}_0)\big) + \epsilon \times \sum_{i=1}^F |w_i| \,.$$

When the current epoch $e < E_w$, the training dataset is used to select the best features:

$$\hat{w} \in \underset{w \in [0,1]^F}{\text{argmin}}\left( \underset{\theta \in \Theta(\alpha,\lambda)}{\min}\left( \tilde{\ell}\big(f_\theta^{\alpha,\lambda}, w, (X_{train}w, Y_{train})\big)\right)\right) \,.$$

As $e = E_w$ is reached, $X_{train}\hat{w}$ is converted to $\hat{X}_{train}$ and optimize the model weights during the last epochs:

$$\hat{\theta} \in \underset{\theta \in \Theta(\alpha,\lambda)}{\text{argmin}}\left( \ell_{MSE}\big(f_\theta^{\alpha,\lambda}, (\hat{X}_{train}, Y_{train})\big)\right) \,.$$

The forecast error with the DNN parameterized by $\hat{\theta}$ on $\mathcal{D}_{valid}$ is used to assess the performance of the selected $\alpha$ and $\lambda$. The architecture and hyperparameters are optimized using:

$$(\hat{\alpha}, \hat{\lambda}) \in \underset{\alpha \in \mathcal{A}}{\text{argmin}}\left( \underset{\lambda \in \lambda(\alpha)}{\text{argmin}}\left( \ell_{MSE}\big(f_{\hat{\theta}}^{\alpha,\lambda}, (X_{valid}\hat{w}, Y_{valid})\big)\right)\right) \,.$$

Finally, the framework output is $\ell_{MAPE}$, the Mean Absolute Percentage Error. Given a load series $Y = (\mathbf{y}_1 \ldots \mathbf{y}_n)$ and the predictions $\hat{Y} = (\hat{y}_1, \ldots \hat{y}_n$, $\text{MAPE}(Y, \hat{Y}) = 1/n \sum_{i=1}^n |(\mathbf{y}_i - \hat{y}_i)/\mathbf{y}_i|$. The MAPE is computed using the DNN with the best architecture, hyperparameters, weights and features on the test dataset:

$$\ell_{MAPE}\big(f_{\hat{\theta}}^{\hat{\alpha},\hat{\lambda}}, (X_{test}\hat{w}, Y_{test})\big) \,.$$

Keisler et al. (2024) noticed that their DNNs were quite unstable. To fix this, our DNNs are trained with a cyclic learning rate, as suggested by Huang et al. (2017), during the second part of DNN training (when $E_w < e \le E_w + E_\theta = E$). When the learning rate is low, the neural network reaches a local minimum. We store its weights at that moment, creating an intermediate model. Immediately after that, the learning rate increases again, bringing the model out of the local minimum. At the end of training, the forecasts of the best intermediate models are averaged.

### 3.3 Meta-Architecture

Each DNN $f \in \Omega$ should map an input $X \in \mathbb{R}^{b \times H \times F}$ into a target $Y \in \mathbb{R}^{b \times H}$, where $b$ represents the size of the batch. The generic search space defined Section 3.1 should be restrained to architectures that map a two-dimensional input to a one-dimensional output. Therefore, each model $f^{\alpha,\lambda} \in \Omega$ consists in two DAGs $\Gamma_1$ and $\Gamma_2$. The graph $\Gamma_1$ is made of 2-dimensional layers operations to treat the matrix $X$ and is parameterized by $\alpha_1$ and $\lambda_1$. A flattened layer follows $\Gamma_1$ to transform the 2-dimensional latent representation into a 1-dimensional one. The graph $\Gamma_2$ is then made of 1-dimensional layers operations and is parameterized by $\alpha_2$ and $\lambda_2$. We have $\alpha = [\alpha_1, \alpha_2]$ and $\lambda = [\lambda_1, \lambda_2]$. A final output layer maps the output shape of $\Gamma_2$ to $H$. The operations that can be chosen at the nodes of $\Gamma_1$ and $\Gamma_2$ are given Section C. A representation of the meta-model is displayed in Figure 2.

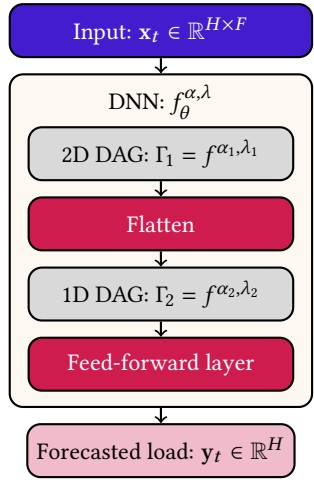

Figure 2: Daily meta-model for load datasets.

### 3.4 Search Algorithm

For the purposes of this article, we have implemented an asynchronous (or steady-state) evolutionary algorithm (SSEA) as our search algorithm. However, the framework is implemented so that other search algorithms can be used. In our experiments (see Section 4), SSEA is compared with simple random search (RS). Training a DNN is very expensive in terms of time and computational resources. We have access to HPC (High-Performance Computing) resources and exploit them by modifying the evolutionary algorithm used by Keisler et al. (2024) into a steady-state version (see Liu et al. (2018a)). At the beginning of the algorithm, a set of $K$ random DNNs is generated. They will all have small architectures with a small number of layers $m$. The idea is to start the optimization with simple DNNs to reduce the chance of ending up with overly complex, heavy and unstable DNNs. The weights of the initial features $w$ can be initialized to uniform random values, zeros, or ones. Then, each initial solution is trained and evalauted on $\mathcal{D}_{train}$ and $\mathcal{D}_{valid}$ to create our population of size $K$. For a certain number of iterations $B$, once a processus is free, two solutions from our population are chosen using a tournament selection. We use the crossover and mutation operators suggested by Keisler et al. (2024) to create two offspring that would be trained and evaluated by the free process. Then, for each offspring, if its loss $\ell_{MAPE}$ is less than the worst loss from the population, the offspring replaces the worst individual. Using an asynchronous version instead of the classical one avoids waiting for a whole generation to be evaluated and saves some time.

## 4 Experiments

In this section, EnergyDragon performance are evaluated on the French load from March 2019 to March 2020, just before the first lockdown. We have used a rather old year because it is the last year with a stable regime. Since this year, the French load has experienced large perturbations during the COVID lockdowns or the energy crisis. Comparing the performance of steady-state models, which is the subject of this article, over periods that are too volatile, without a re-calibration mechanism, is irrelevant. This issue is further discussed Section 5.

### 4.1 Dataset

The dataset comes from the website of the French Transmission System Operator[3] (RTE) and contains the French national load data at half-hourly intervals. Therefore, each day contains $H = 48$

---

[3]https://www.rte-france.com/eco2mix

time steps. We trained our models from March 2015 to March 2019 and compare the performance from March 2019 to March 2020 using the MAPE ($\ell_{MAPE}$ ). For this dataset thirty-four explanatory variables can be used. The weather data contains the national temperature along with exponential smoothing variants of parameters going from 0.7 to 0.998, wind and cloud cover. Calendar features include the day of the week, the month, the year, if the day correspond to a public holidays or a surrounding day.

## 4.2 Baseline

We compare our results to models at the state-of-the-art in load forecasting: a Generalized Additive Model (GAM) used in the industry, the CNN/MLP model and to two AutoML/AutoDL approaches: AutoPytorch (Zimmer et al., 2021) and a version of DARTS (Liu et al., 2018b) applied on the hand-crafted DNN. AutoPytorch includes the hyperparameters tuning, model selection and ensembling of simple regression models such as Random Forest, Support Vector Machine (SVM) or Catboost for example, therefore they are not directly included in the baseline.

**Generalized Additive Model**. The GAMs are state of the art for load forecasting (see among others Pierrot and Goude (2011) or Wood (2017)). The output $Y$ is explained as the sum of smooth functions of the explanatory variables: $Y = g_1(X_1) + g_2(X_2) + g_3(X_3, X_4) + \ldots$ where the $g_i$ are linear or a special type of piecewise polynomial functions called splines. The GAM developed for this use case is instantaneous, meaning that a model is fitted for each of the 48 time steps of the day. For our experiments, we chose a GAM that is used in the industry and therefore cannot reveal its explicit formula. The GAM takes about twenty explanatory variables as input.

**Deep Neural Networks**. We included our CNN/MLP inspired by Farsi et al. (2021) and He (2017) in our baseline. Unlike the GAM, a single model is used to predict the 48 time steps of the day. The input variables are divided into two groups of about ten features. One group is processed by parallel one-dimensional convolutions and the other by feed-forward layers. The branches are then concatenated and processed by more feed-forward layers. The detailed architecture can be found Section A.1. The model was trained with the Adam optimizer (Kingma and Ba, 2017) during 500 epochs.

**AutoPytorch**. We used the tabular regression API of AutoPytorch[4]. This framework combines an AutoML pipeline for traditional regression models (e.g., RandomForest, CatBoost or LightGBM) with the tuning of DNNs. The search-space used for the AutoDL part is made of MLPs, residual connections and Normalization Layers. The framework does not allow us to map two-dimensional inputs to one-dimensional targets, so each moment of the day was forecasted independently. The hour of the day and the instant were added as explanatory variables. For a fair comparison, the same global optimization budget was set for both AutoPytorch and EnergyDragon: 24 hours, and the same budget by model: 15 minutes. Two versions are used in the baseline: with the traditional regression baseline and with only the AutoDL part (fairer with EnergyDragon).

**DARTS**. We optimized the CNN/MLP architecture using the DARTS (Liu et al., 2018b) algorithm. The search space is described in detail in Section B. In short, parts of the original architecture are replaced with DARTS cells. Each cell is a DAG where the links represent candidate operations. During optimization, multiple operations are considered for each link and are associated with a probability of being selected. These probabilities are optimized by gradient descent. At the end of the optimization, the operation with the highest probability is chosen in the final architecture. The model weights and the operation probabilities are optimized alternatively, with 500 epochs for the weights and 200 epochs for the probabilities, using the Adam optimizer Kingma and Ba (2017).

---

[4]`https://github.com/automl/Auto-PyTorch/tree/master`

**EnergyDragon.** For EnergyDragon (hereafter called ED),the global time budget is fixed to 24 hours. The features are optimized during 500 epochs and the weights during 200 epochs. For the steady-state evolutionary algorithms, the initial population size is set to $K = 100$. A DNN cannot be trained for more than 15 minutes. The baseline compared five versions of ED. One with a random search algorithm, called ED RS, the other versions use the steady-state evolutionary algorithm. ED SSEA is implemented with the mutation operators of DRAGON but without the crossover, and ED SSEA Crossover uses the crossover. Finally, ED SSEA CNN/MLP and ED SSEA Crossover CNN/MLP include the MLP/CNN model in the initial population. This means that 99 models are randomly initialized and the remaining one is the CNN/MLP model.

## 4.3 Results

| Model | MAPE | RMSE (in MW) |
|---|---|---|
| GAM | 1.398% | 929.8 |
| AutoPytorch | 17.999% | 10641.7 |
| AutoPytorch with the traditional baseline | 2.022% | 1243.2 |
| CNN/MLP (handcrafted DNN) | 1.721% | 1164.6 |
| DARTS | 1.600% | 1085.6 |
| ED RS | 1.374% | 902.3 |
| ED SSEA | 1.258% | 851.4 |
| ED SSEA Crossover | 1.190% | 837.8 |
| ED SSEA Crossover CNN/MLP | 1.182% | 816.3 |
| ED SSEA CNN/MLP | **1.131%** | **803.4** |

Table 1: MAPE and RMSE of the different models from our baseline. The reference model is the GAM and the best model is highlighted in bold.

We evaluated each algorithm from the baseline on the French load signal. Each version from ED was run using 20 GPUs V100, and AutoPytorch using 2 Quadro RTX 6000 which are faster than the V100. We used in total approximatively 336 GPU-hours. Each algorithm was run with a global seed of 0 to ensure reproducibility. The results can be found in Table 1. In addition to the MAPE function, the Root Mean Squared Error is (RMSE) is also reported. For all proposed versions, the results of the EnergyDragon algorithms beat all other models from the baseline. The AutoPytorch framework had the worst results, even with the traditional models. The lack of feature selection may explain these results. Our CNN/MLP handcrafted model was slightly improved by the DARTS framework, but both versions cannot compete with the reference model (GAM). Among the ED results, the random search got the worst results, which demonstrates the performance of our search algorithm (see D.3 for convergence plots). Although the CNN/MLP model does not outperform the GAM, it is still useful to use it as an input for ED. In fact, both versions with and without crossover with the CNN/MLP in the initial population outperformed the versions without. Finally, the crossover helped to improve the performance of ED without the CNN/MLP as input, but the best version of ED was with the CNN/MLP and without the crossover. The initial population of this last algorithm already contains a good candidate (the CNN/MLP model), and therefore does not need as much exploration (with the crossover) as the version without the CNN/MLP. The models found by EnergyDragon for each setup can be found in Section D.1. The best version of EnergyDragon: ED SSEA CNN/MLP improves the predictions of GAM by 19%. Figure 3 shows the forecast of GAM and ED SSEA CNN/MLP for the last week of November. Forecasts from other algorithms can be found in Section D.2. The forecast signals have similar shapes for GAM and ED SSEA CNN/MLP,

but GAM has a larger bias and overpredicts the load. More details on the experimental results can be found in Section D, as well as another case study on the Norwegian load Section E.

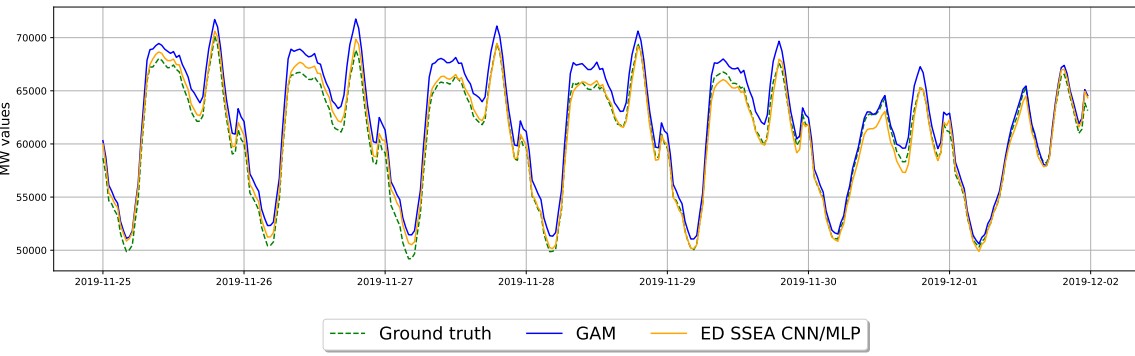

Figure 3: Load power forecasting for the last week of November 2019. The ground truth is displayed in dotted line, the GAM forecast is drawn with a blue line whereas the forecast from the best version of EnergyDragon (ED SSEA CNN/MLP) is drawn in yellow.

## 5 Conclusion

This paper explains how we applied AutoDL to a real-world application: load forecasting. The existing works in the AutoDL community were not sufficient to be used directly in our case, and we had to develop our own AutoDL framework, EnergyDragon. This framework is able to automatically select input features and optimize DNN architectures and hyperparameters to generate performing models. We demonstrate on the French load signal that EnergyDragon is able to outperform a state-of-the-art model in load forecasting: a generalized additive model used in the industry as well as an AutoML framework designed for tabular regression: AutoPytorch. Future work should focus on automatically re-calibrating the models found by EnergyDragon so that they can be used for short-term forecasting in rather erratic periods. In addition, the industry requires the interpretability of the forecasting models. DNNs are known to be black boxes, and to be accepted as an industrial solution, we will have to work on finding ways to interpret their forecasts.

## 6 Broader Impact Statement

EnergyDragon automates the creation of powerful load forecasting models. It could be used for load forecasting in different countries, which could be useful in the fight against climate change. In fact, meeting the carbon neutrality targets of the Paris Agreement depends for many countries on increasing the share of electricity in their energy mix through the massive deployment of renewable energy. Load forecasting will be critical to managing these future power grids, which will include many intermittent energy sources. Accurate load forecasts, combined with forecasts of renewable energy production, will enable the activation of electricity flexibility and reduce the use of high carbon-emitting peak generation resources. Smarter, more flexible power grids that can accommodate large amounts of renewable energy are cited by Foley et al. (2020) as one of the most important solutions to climate change. However, it should be noted that the non-interpretability of DNNs can pose a risk to power system operators. It is difficult to predict in advance how they will react to unknown situations and how to correct their load forecasts in such situations, which could lead to operational disruptions or financial losses.

# References

**Acknowledgements.**

Chen, D., Chen, L., Shang, Z., Zhang, Y., Wen, B., and Yang, C. (2021). Scale-Aware Neural Architecture Search for Multivariate Time Series Forecasting. *arXiv:2112.07459 [cs]*. arXiv: 2112.07459.

Cordonnier, J., Loukas, A., and Jaggi, M. (2019). On the relationship between self-attention and convolutional layers. *CoRR*, abs/1911.03584.

Dai, Z., Yang, Z., Yang, Y., Carbonell, J. G., Le, Q. V., and Salakhutdinov, R. (2019). Transformer-xl: Attentive language models beyond a fixed-length context. *CoRR*, abs/1901.02860.

del Real, A. J., Dorado, F., and Durán, J. (2020). Energy Demand Forecasting Using Deep Learning: Application to the French Grid. preprint, ENGINEERING.

Deng, D., Karl, F., Hutter, F., Bischl, B., and Lindauer, M. (2022). Efficient automated deep learning for time series forecasting. *arXiv preprint arXiv:2205.05511*.

Farrokhabadi, M., Browell, J., Wang, Y., Makonin, S., Su, W., and Zareipour, H. (2022). Day-ahead electricity demand forecasting competition: Post-covid paradigm. *IEEE Open Access Journal of Power and Energy*, 9:185–191.

Farsi, B., Amayri, M., Bouguila, N., and Eicker, U. (2021). On Short-Term Load Forecasting Using Machine Learning Techniques and a Novel Parallel Deep LSTM-CNN Approach. *IEEE Access*, 9:31191–31212.

Foley, J., Wilkinson, K., Frischmann, C., Allard, R., Gouveia, J., Bayuk, K., Mehra, M., Toensmeier, E., Forest, C., Daya, T., Gentry, D., Myhre, S., Mukkavilli, s. K., Yussuff, A., Mangotra, A., Metz, P., Wartenberg, A., Anand, C., Jafary, M., and Rodriguez, B. (2020). *The Drawdown Review (2020) - Climate Solutions for a New Decade*.

Gaillard, P. (2015). *Contributions à l'agrégation séquentielle robuste d'experts : travaux sur l'erreur d'approximation et la prévision en loi. Applications à la prévision pour les marchés de l'énergie*. PhD thesis, Université Paris-Sud 11.

Grinsztajn, L., Oyallon, E., and Varoquaux, G. (2022). Why do tree-based models still outperform deep learning on tabular data? *arXiv preprint arXiv:2207.08815*.

He, W. (2017). Load Forecasting via Deep Neural Networks. *Procedia Computer Science*, 122:308–314.

Huang, G., Li, Y., Pleiss, G., Liu, Z., Hopcroft, J. E., and Weinberger, K. Q. (2017). Snapshot ensembles: Train 1, get m for free. *arXiv preprint arXiv:1704.00109*.

Hutter, F., Kotthoff, L., and Vanschoren, J. (2019). *Automated machine learning: methods, systems, challenges*. Springer Nature.

Keisler, J., Talbi, E.-G., Claudel, S., and Cabriel, G. (2024). An algorithmic framework for the optimization of deep neural networks architectures and hyperparameters. *Journal of Machine Learning Research*, 25(201):1–33.

Kingma, D. P. and Ba, J. (2017). Adam: A method for stochastic optimization.

Li, J., Cheng, K., Wang, S., Morstatter, F., Trevino, R. P., Tang, J., and Liu, H. (2017). Feature selection: A data perspective. *ACM computing surveys (CSUR)*, 50(6):1–45.

Liu, H., Simonyan, K., Vinyals, O., Fernando, C., and Kavukcuoglu, K. (2018a). Hierarchical representations for efficient architecture search.

Liu, H., Simonyan, K., and Yang, Y. (2018b). Darts: Differentiable architecture search. *arXiv preprint arXiv:1806.09055*.

L'Heureux, A., Grolinger, K., and Capretz, M. A. (2022). Transformer-based model for electrical load forecasting. *Energies*, 15(14):4993.

Mamun, A. A., Hoq, M., Hossain, E., and Bayindir, R. (2019). A Hybrid Deep Learning Model with Evolutionary Algorithm for Short-Term Load Forecasting. In *2019 8th International Conference on Renewable Energy Research and Applications (ICRERA)*, pages 886–891, Brasov, Romania. IEEE.

Nedellec, R., Cugliari, J., and Goude, Y. (2014). Gefcom2012: Electric load forecasting and backcasting with semi-parametric models. *International Journal of forecasting*, 30(2):375–381.

Novaes, A. L. F., Araujo, R. A. d. M., Figueiredo, J., and Pavanelli, L. A. (2021). A New State-of-the-Art Transformers-Based Load Forecaster on the Smart Grid Domain. *arXiv:2108.02628 [cs]*. arXiv: 2108.02628.

Pierrot, A. and Goude, Y. (2011). Short-term electricity load forecasting with generalized additive models. *Proceedings of ISAP power*, 2011.

Rahman, A., Srikumar, V., and Smith, A. D. (2018). Predicting electricity consumption for commercial and residential buildings using deep recurrent neural networks. *Applied Energy*, 212:372–385.

Sehovac, L. and Grolinger, K. (2020). Deep Learning for Load Forecasting: Sequence to Sequence Recurrent Neural Networks With Attention. *IEEE Access*, 8:36411–36426.

Tibshirani, R. (1996). Regression shrinkage and selection via the lasso. *Journal of the Royal Statistical Society Series B: Statistical Methodology*, 58(1):267–288.

Vaswani, A., Shazeer, N., Parmar, N., Uszkoreit, J., Jones, L., Gomez, A. N., Kaiser, Ł., and Polosukhin, I. (2017). Attention is all you need. *Advances in neural information processing systems*, 30.

Vilmarest, J. D. (2022). *State-Space Models for Time Series Forecasting. Application to the Electricity Markets. (Modèles espace-état pour la prévision de séries temporelles. Application aux marchés électriques)*. PhD thesis, Sorbonne University, Paris, France.

Wood, S. N. (2017). *Generalized additive models: an introduction with R*. CRC press.

Zhou, H., Zhang, S., Peng, J., Zhang, S., Li, J., Xiong, H., and Zhang, W. (2021a). Informer: Beyond Efficient Transformer for Long Sequence Time-Series Forecasting. *arXiv:2012.07436 [cs]*. arXiv: 2012.07436.

Zhou, H., Zhang, S., Peng, J., Zhang, S., Li, J., Xiong, H., and Zhang, W. (2021b). Informer: Beyond efficient transformer for long sequence time-series forecasting. In *Proceedings of the AAAI Conference on Artificial Intelligence*, volume 35, pages 11106–11115.

Zimmer, L., Lindauer, M., and Hutter, F. (2021). Auto-pytorch: Multi-fidelity metalearning for efficient and robust autodl. *IEEE Transactions on Pattern Analysis and Machine Intelligence*, 43(9):3079–3090.


## A Handcrafted DNN for load forecasting: the CNN/MLP model

### A.1 Architecture

Figure 4 shows the CNN/MLP architecture we have handcrafted for load forecasting. As described in Section 4, the model has two inputs with different features. One is handled by two parallel convolutions and the other by an MLP. The three branches are then concatenated and fed into another MLP layer. The input data is a bit different than in EnergyDragon, where $X_{\text{EnergyDragon}} \in \mathbb{R}^{H \times F}$. In the CNN/MLP model, the features are split into $X_{\text{Conv}}$ and $X_{\text{Feed}}$, which contain $F_C$ and $F_F$ features, respectively, so that $F_C + F_F = F$. Within $X_{\text{Conv}}$ and $X_{\text{Feed}}$, the features are concatenated into a one-dimensional vector, i.e. $X_{\text{Conv}} \in \mathbb{R}^{HF_C}$ and $X_{\text{Feed}} \in \mathbb{R}^{HF_F}$.

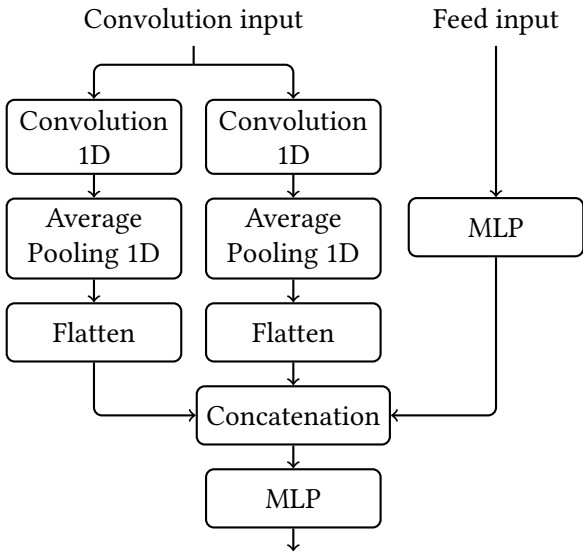

Figure 4: CNN/MLP Architecture

To represent the CNN/MLP model in EnergyDragon and to include it in the initial population, we had to make some adjustments. The input data is two-dimensional: $X_{\text{EnergyDragon}} \in \mathbb{R}^{H \times F}$, so we set the first DAG $\Gamma_1$ to an identity layer and we apply the three branches to the flattened data. The architecture of this new model is shown in Figure 5. The architecture shown with EnergyDragon gives slightly worse results than the original one.

### A.2 Self-attention

Before creating a fully automated framework for finding neural networks for load forecasting, we searched for DNNs that might be interesting for our problem. The Transformer model (Vaswani et al., 2017), which has recently achieved state-of-the-art results in several areas, naturally caught our attention. We tried to use it on our problem, but without much success: we tried different hyperparameters and the load prediction embedding from the Informer (Zhou et al., 2021b), but we could not go below 4% of MAPE. However, one of the major innovations of the Transformer is the self-attention layer. In the vanilla Transformer model, the attention layer is position-invariant, meaning that no assumption is made about the order of the inputs, and the permutation of the input data does not change the result. Therefore, in the original Transformer (Vaswani et al., 2017), the absolute position $P_{data}$ of the data $X_{data}$ in the data set is added to the input data: $X = X_{data} + P_{data}$. The attention scores can then be defined as:

$$A = X W_Q W_K^T X^T = (X_{data} + P_{data}) W_Q W_K^T (X_{data} + P_{data})^T ,$$

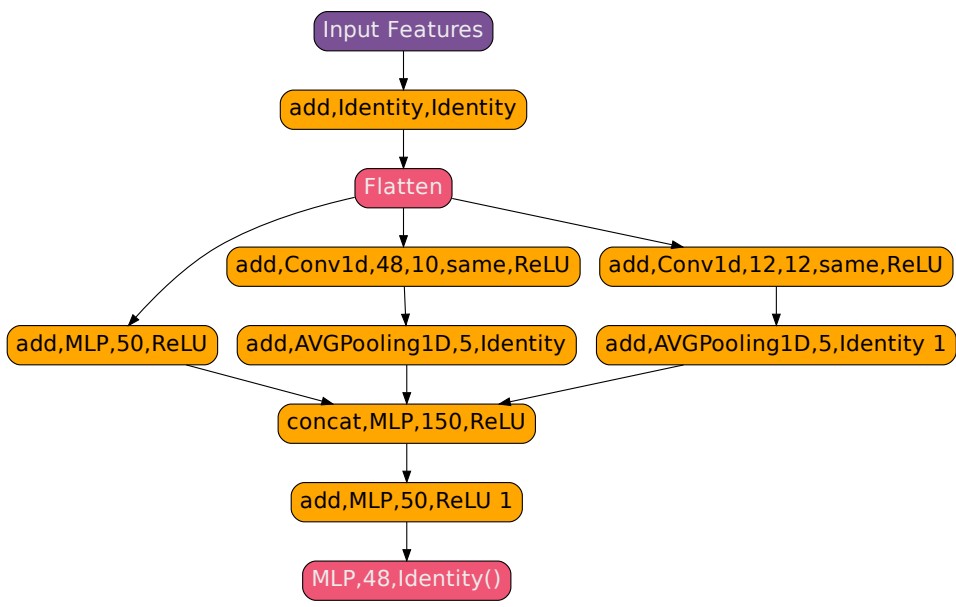

Figure 5: CNN/MLP Architecture represented with EnergyDragon.

where $W_Q$ and $W_K$ are the query and key weights matrices from the original self-attention.

Later, Dai et al. (2019) introduced relative encoding in their Transformer XL. The idea is to consider only the position difference between query and key instead of the absolute position. They redefined the attention score between a query $x_q$ and a key $x_k \in X_{data}$:

$$A_{q,k} = x_q^T W_Q^T W_K x_k + x_q^T W_Q^T \hat{W}_K r_{k-q} + u^T W_K x_k + v^T \hat{W}_K r_{k-q},$$

where $r_{k-q}$ is a coding of the relative position, $u$ and $v$ are new attention parameters optimized by backpropagation. From this new attention formulation, Cordonnier et al. (2019) proved that by setting some conditions, the attention layer can be forced to learn as a 2D convolutional layer.

We implemented the attention layers as defined by Cordonnier et al. (2019) for one- and two-dimensional data, and set a parameter "initialization" to indicate whether the layer weights should be initialized to perform a convolution, or if they should be randomized. We replaced the convolutions from the CNN/MLP architecture with these self-attention layers, and compared the performance of three different models: the original CNN/MLP architecture, a self-attention/MLP architecture with the self-attention weights initialized as a convolution, and a self-attention/MLP architecture with randomized self-attention weights. We compared the performance of these models on our data for 10 different seeds. The results are shown in Figure 6. The models with self-attention layers perform better than the original model, reducing the average MAPE from 1.60% for the original model to 1.51% and 1.48% for the convolution and random initializations, respectively. This last model even reached a MAPE of 1.40%. We included this self-attention layer in the search spaces of DARTS and EnergyDragon as detailed in Sections B and C.

## B DARTS

Differential Architecture Search, also called DARTS, was introduced by Liu et al. (2018b), originally for computer vision and NLP tasks. The cell-based search space is composed of either stacked cells to form a convolutional network, or recursively connected cells to form a recurrent network. Each

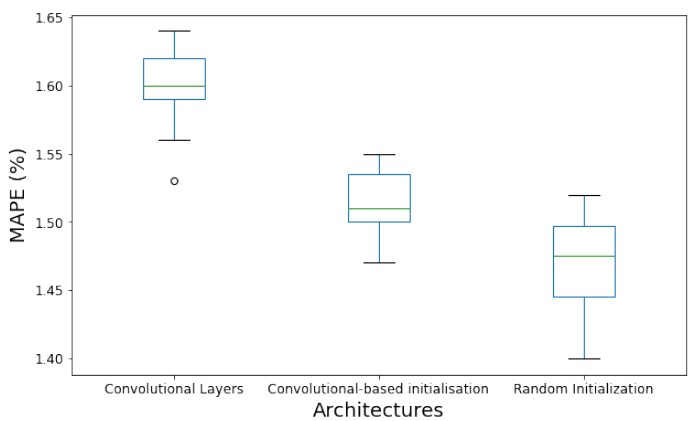

Figure 6: MAPE on the RTE dataset for three versions of the CNN/MLP model trained with 10 different seeds.

cell is defined as a DAG with $N$ nodes $x^{(i)}$ (latent representations) and edges $O^{(i,j)}$ connecting the nodes (operations). Each DAG has two input nodes and one output. For convolutional cells, the input nodes are the outputs of the two previous cells. For recurrent cells, the input nodes are the input at the current step and the state carried over from the previous step. The cell output is obtained by concatenating all intermediate nodes. An intermediate node is defined as the sum of all previous latent representations after a candidate operation: $x^{(j)} = \sum_{i<j} O^{(i,j)}(x^{(i)})$. The goal of DARTS is to find the best operations between each node. The main idea introduced by DARTS is the relaxation of the search space. A set of candidate operations $\mathcal{O}$ (e.g., convolution, linear, or identity layers) is associated with each connection. Each candidate $o \in \mathcal{O}$ is assigned a probability parametrized by a real $\alpha_o \in \mathbb{R}$ of being part of the final architecture. The relaxed operation between the nodes $i$ and $j$ can be defined as :

$$\bar{o}^{(i,j)} = \sum_{o \in \mathcal{O}} \frac{exp(\alpha_o^{(i,j)})}{\sum_{o' \in \mathcal{O}} exp(\alpha_{o'}^{(i,j)})} o^{(i,j)} .$$

At the beginning of the search, the parameters are uniformly initialized for all candidate operations. Then, during model training, the $\alpha_o$ and the network weights are updated alternately using gradient descent. Finally, an argmax function is used to select the candidate operation with the higher probability to build the final architecture. DARTS is a popular framework in the NAS community because it is easy to implement and fast compared to other optimization techniques. It does not require training all new solutions picked from the search space from scratch. The final architecture is a subgraph of the meta-architecture. A drawback of this method is that there is no theoretical guarantee that the optimal subgraph of an optimal meta-architecture is an optimal solution.

Inspired by the work of Chen et al. (2021), where DARTS is used to optimize a DNN for multivariate time series forecasting, we applied DARTS to optimize the CNN/MLP architecture. As in the CNN/MLP model, the general structure of the search space (see Figure 7a) consists of two inputs. Each input is handled by a dedicated DARTS cell. The *Conv* cell (see figure 7b) replaces the two convolutional branches of the CNN/MLP model. The first *Feed* cell replaces the first MLP branch and the Second replaces the last MLP branch. Each cell has a maximum of $N_c = 4$ nodes and the candidate operations $o_{i,j}$ on each connection are those from the original architecture: e.g., Conv1d/Pooling layers for the Conv cell (see Figure 7b) or MLP/Identity layers for the Feed cells (see Figure 7c). We also added zero operations and Attention layers as defined in Section A.2 to our search space, with the same hyperparameters as the convolutional layers.

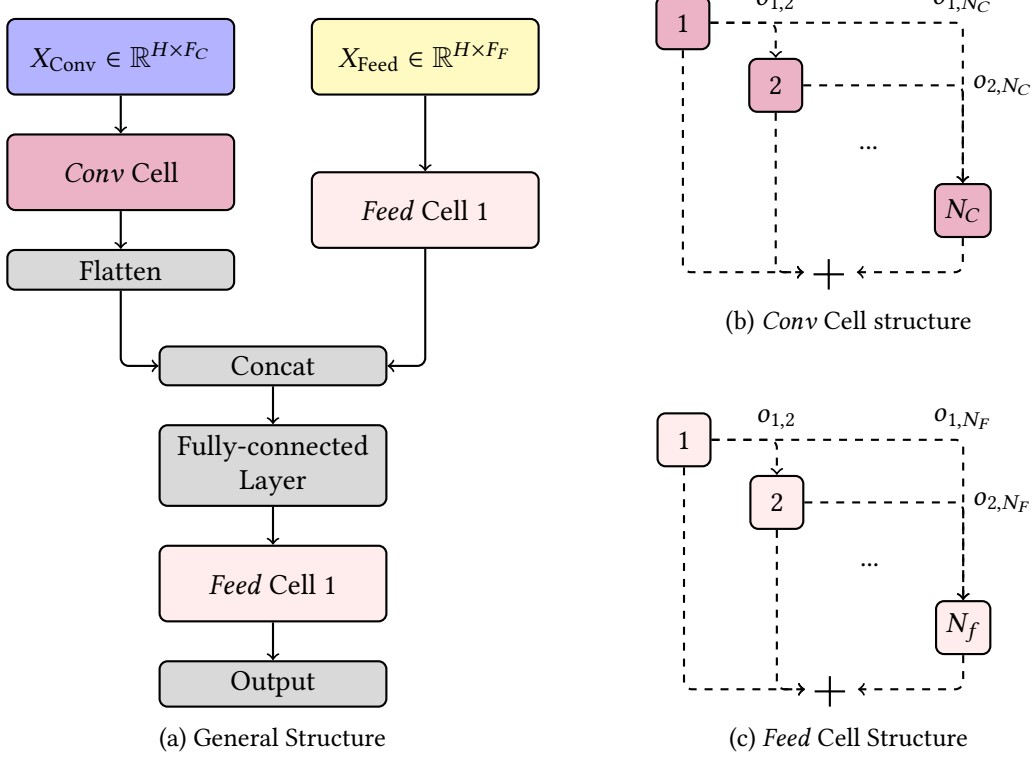

Figure 7: DARTS Search Space for load forecasting.

## C  EnergyDragon search space details

The layers (operations) used in our search space are detailed Table 2. Most layers are adapted or used in both $\Gamma_1$ (two-dimensional data) and $\Gamma_2$ (one-dimensional data), except for the Temporal Attention and the Spatial Attention, which are specific to $\Gamma_1$. Given an input data $X \in \mathbb{R}^{H \times f \times d}$, where $f$ and $d$ would be two latent dimensions within the DNN, the attention matrix is the same as defined Section A.2:

$$\text{Attention}(X) = \text{softmax}(XW_Q W_K^T X^T + XW_Q \hat{W}_K^T \delta_R + uW_K^T X^T + v\hat{W}_k^T \delta_R)XW_O + b_O,$$

with $W_Q$ and $W_K$ the query and key weight matrices. In the Temporal Attention case, $W_Q, W_K \in \mathbb{R}^{H \times d \times N_h \times 2}$ and in the Spatial attention case, $W_Q, W_K \in \mathbb{R}^{f \times d \times N_h \times 2}$, where $N_h \in \mathbb{N}^+$ is the head number. The Temporal Attention computes attention scores between the time steps as depicted Figure 8a, whereas the Spatial Attention computes attention scores between the features, as shown Figure 8b.

## D  Additional experimental results

In the Section we give more information one the experimental results presented Section 4. The Section D.1 detail the different architectures and hyperparameters found by the versions of EnergyDragon presented Section 4. Section D.2 present the forecast of various algorithms from our baseline over the last week of November. The EnergyDragon convergences over time are given Section 15. Finally, Section D.4 discuss the features use by the GAM, the CNN/MLP and the different versions of EnergyDragon.

| Layer type | Optimized hyperparameters | |
|---|---|---|
| Identity | - | |
| Fully-Connected (MLP) | Output shape | Integer |
| Self-Attention (Section A.2) | Operation dimension | [temporal, spatial] ($\Gamma_1$ only) |
| | Initialization type | [convolution, random] |
| | Heads number | Integer |
| | Output dimension | Integer |
| 1D/2D Convolution | Kernel size | Integer |
| | Output dimension | Integer |
| 1D/2D Pooling | Pooling size | Integer |
| | Pooling type | [Max, Average] |
| 1D/2D Normalization | Normalization type | [Batch, Layer] |
| Dropout | Dropout rate | Float |

Table 2: Layers available and their associated hyperparameters in the EnergyDragon search space (for $\Gamma_1$ and $\Gamma_2$).

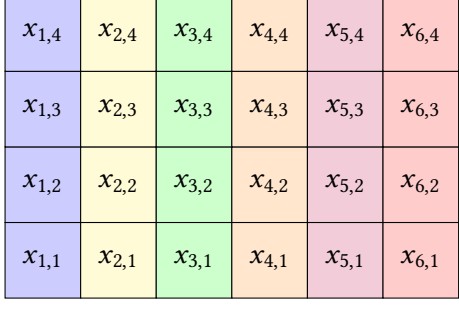

(a) Temporal attention: the attention scores are computed along the time axis. There is no inter-variable interactions

(b) Spatial attention: the attention scores are computed along the feature axis. There is no intra-variable interactions

Figure 8: Spatial and Temporal Attentions vizualisation, applied on $X = \{x_t\}_{t=1}^{H} = \{x_i\}_{i=1}^{F} \in \mathbb{R}^{H \times F}$.

### D.1 Models found by the algorithms

Respectively Figures 9, 10, 11, 12 and 13 represent the architectures and hyperparameters found by ED RS, ED SSEA, ED SSEA Crossover, ED SSEA Crossover CNN/MLP and the best algorithm ED SSEA CNN/MLP. The model found by the Random Search is very simple and based only on the Spatial Attention layer defined in Section C, which computes attention scores between the features. All the found architectures use the attention layers presented in Section A.2. This confirms our first experiments showing that our self-attention layers is efficient for load forecasting. The architectures found by the versions with crossover are more complex, with more layers and connections than the versions without crossover. The appearance of many identity layers invites us to think about pruning our graphs for future work on EnergyDragon. It seems that certain nodes are not necessarily useful, and it might be a good idea to automatically remove them during optimization to avoid ending up with architectures that are too complex or difficult to interpret. The best architecture, which achieves a MAPE of 1.131% as shown in Figure 13, is quite simple.

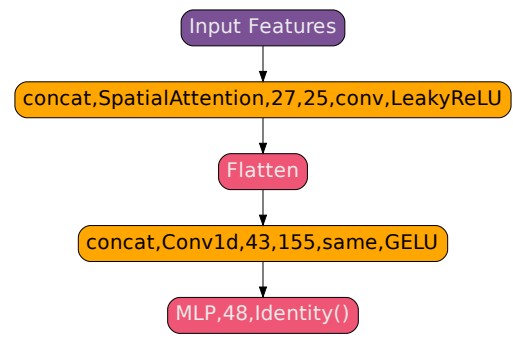

Figure 9: Architecture found by ED RS. Best MAPE=1.374%.

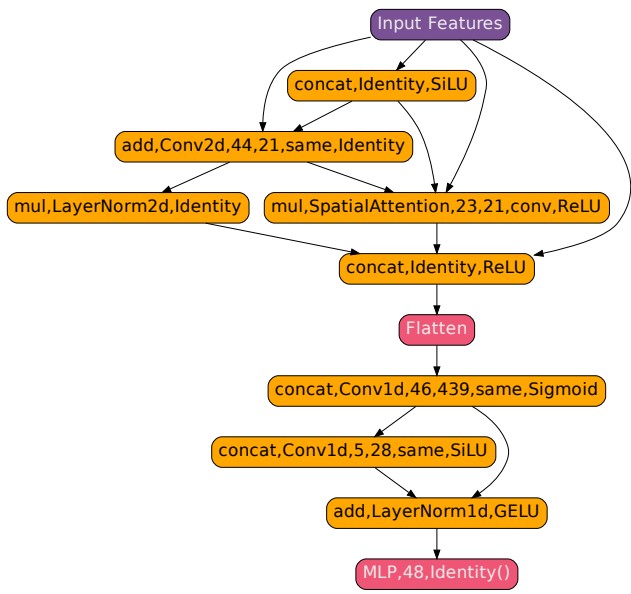

Figure 10: Architecture found by ED SSEA. Best MAPE=1.258%.

Finally, it is difficult to conclude whether including the CNN/MLP as input to EnergyDragon affects the architecture found. Our intuition is that including the CNN/MLP probably has an influence on the selected features. A deeper discussion can be found in Section D.4.

## D.2 Weekly comparative visuals of all baseline forecasts

Forecasts from various models of our baseline for the last week of November can be found in Figure 14. Most models have the same shape and, like GAM, overpredict during this week. AutoPy-torch without the traditional baseline produces a constant signal as shown in Figure 14d, which explains its poor MAPE and RMSE. We compare in Figure 14b the forecast of DARTS with the output of the CNN/MLP model. Using DARTS allowed to improve the overprediction of CNN/MLP, but DARTS is still higher than ED SSEA CNN/MLP as shown in Figure 14c.

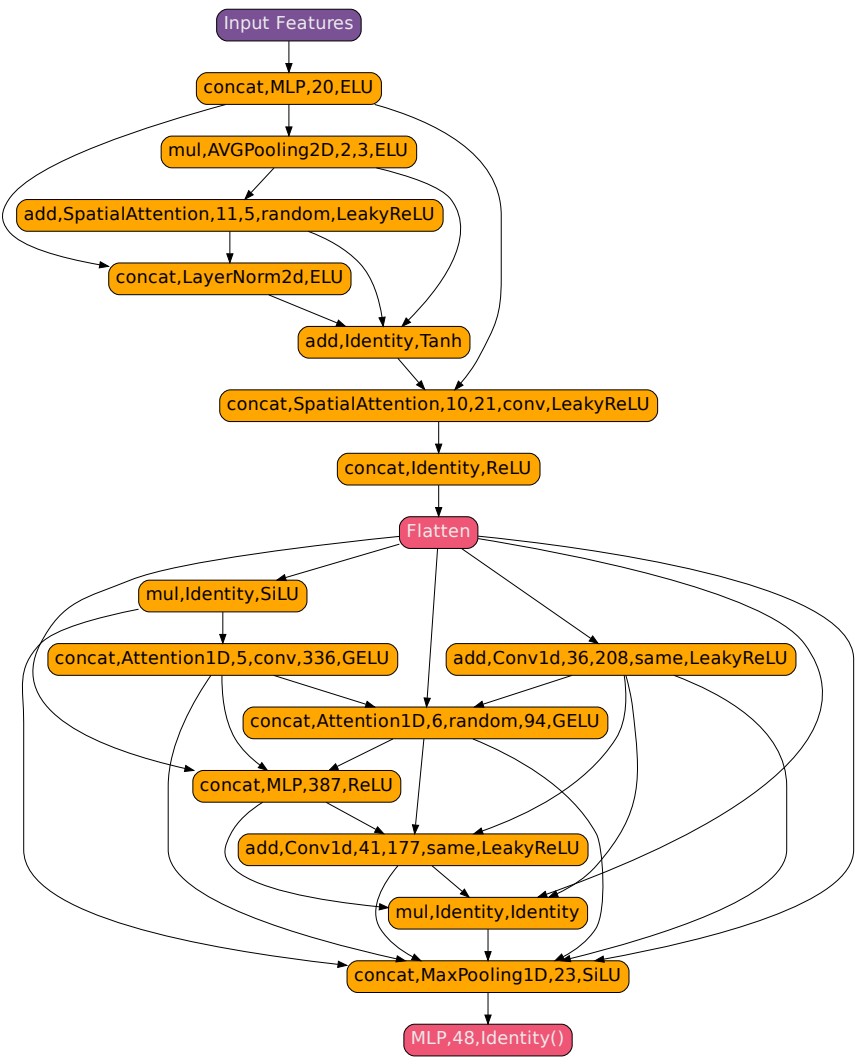

Figure 11: Architecture found by ED SSEA Crossover. Best MAPE=1.190%.

### D.3 EnergyDragon convergence

Figure 15 shows the loss of the best model over time for ED RS, ED SSEA, ED SSEA Crossover, ED SSEA Crossover CNN/MLP, and ED SSEA CNN/MLP. We can see that most versions converge to their best model in less than 10 hours, even if we let the algorithm run for another 10 hours. The versions that include the CNN/MLP in their initial population converge much faster than the version without. ED SSEA CNN/MLP, which gave the best result, converged in a little more than 4 hours.

### D.4 Features

Part of the features used in our experiments cannot be revealed due to industrial confidentiality. Therefore, we have renamed our 33 features from $f_0$ to $f_3$3. Some are weather variables like temperature or wind. Others are general calendar features: the month, the week of the day, or more related to France, like the holidays or the time shift. We present in Figure 16 the features selected by the models. The GAM use very different features compared to the other models, as stated in

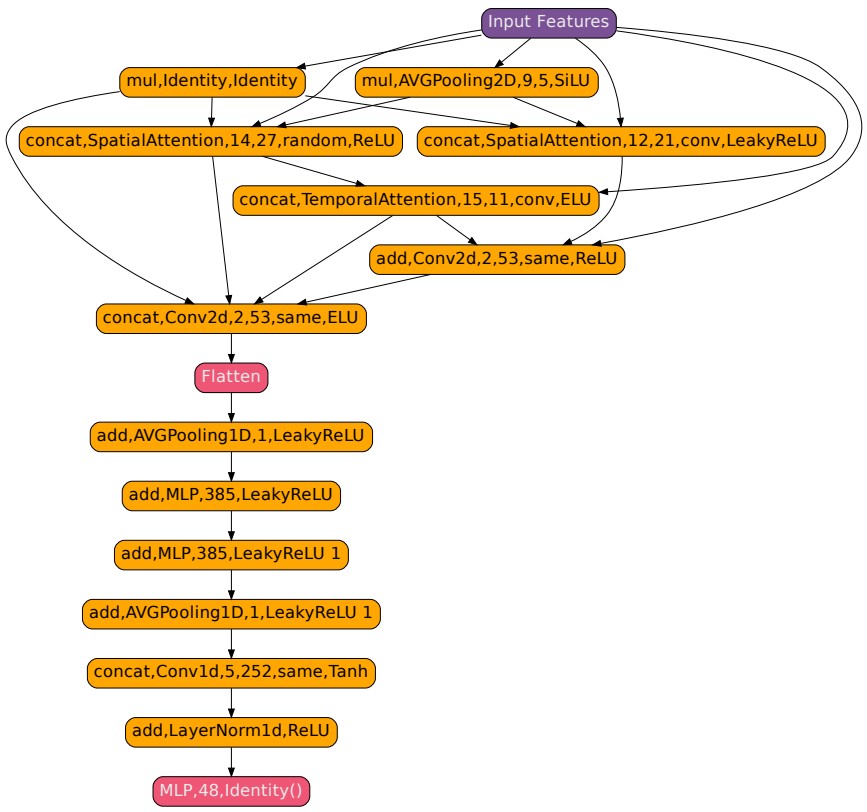

Figure 12: Architecture found by ED SSEA Crossover CNN/MLP. Best MAPE=1.182%.

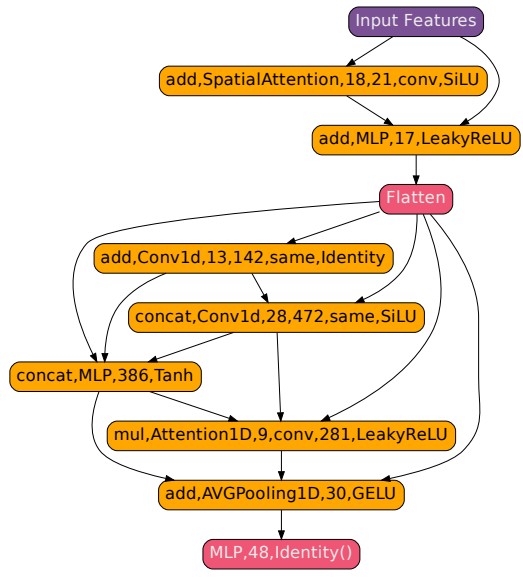

Figure 13: Architecture found by ED SSEA CNN/MLP. Best MAPE=1.131%

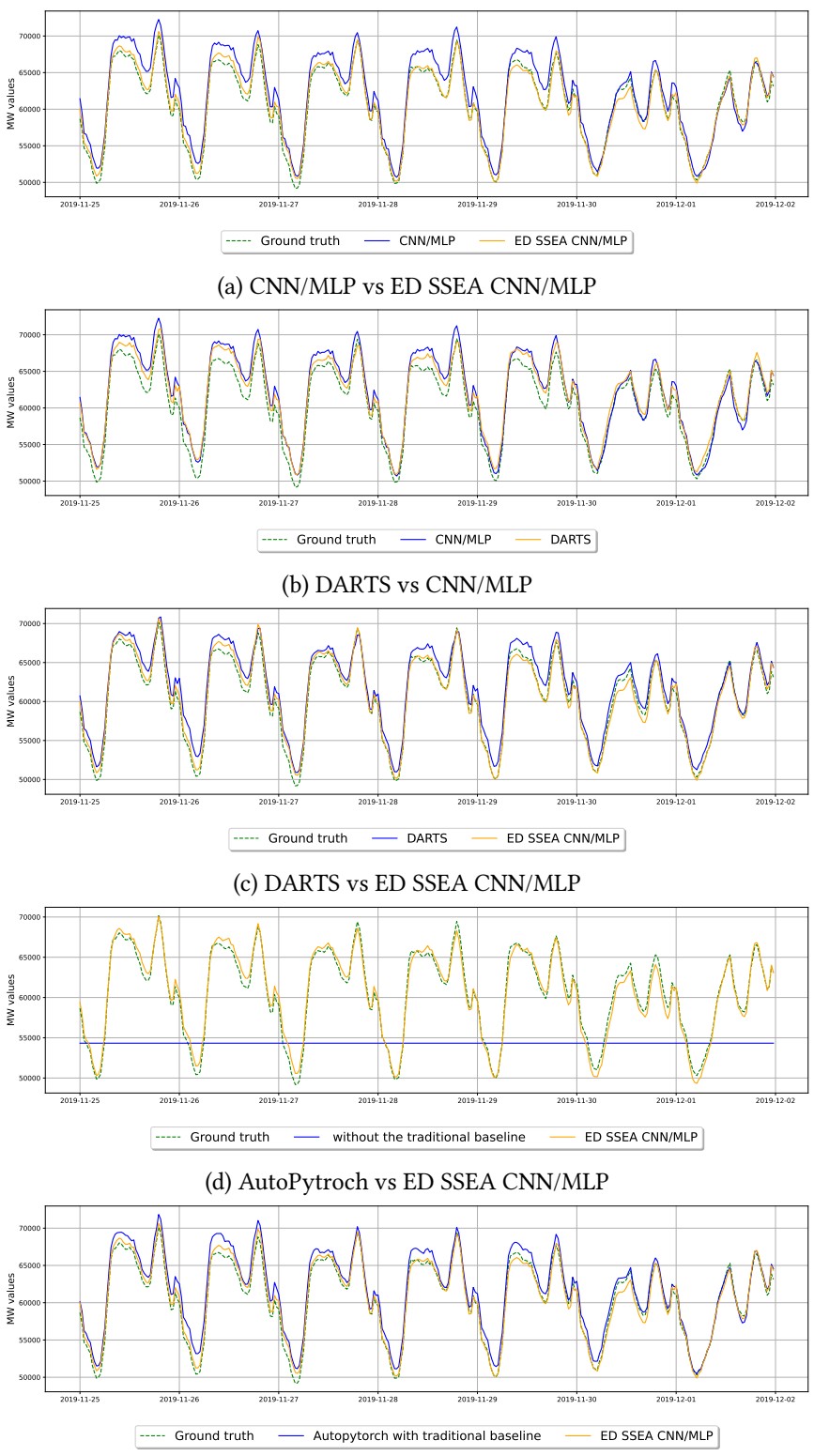

(a) CNN/MLP vs ED SSEA CNN/MLP

(b) DARTS vs CNN/MLP

(c) DARTS vs ED SSEA CNN/MLP

(d) AutoPytroch vs ED SSEA CNN/MLP

(e) AutoPytroch with the traditional baseline vs ED SSEA CNN/MLP

Figure 14: Comparison of the forecasts from various algorithm over the last week of November 2019.

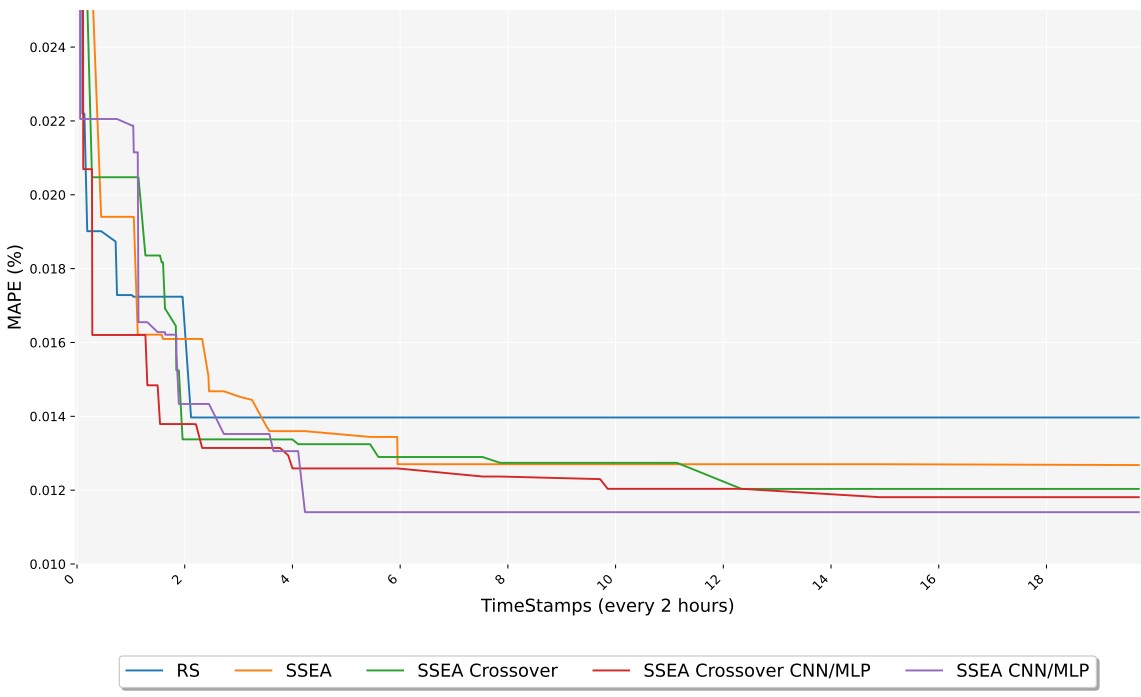

Figure 15: Loss of the best model over time for the different versions of EnergyDragon used in our experiments Section 4.

Section 1. All versions of EnergyDragon selected a number of features close to the number of features used by GAM and CNN/MLP. This means that our LASSO penalization is efficient. Finally, the versions of EnergyDragon with CNN/MLP as input selected a number of features close to the number used by CNN/MLP. As mentioned in Section D.1, this can explain the faster convergence of the version with CNN/MLP as input.

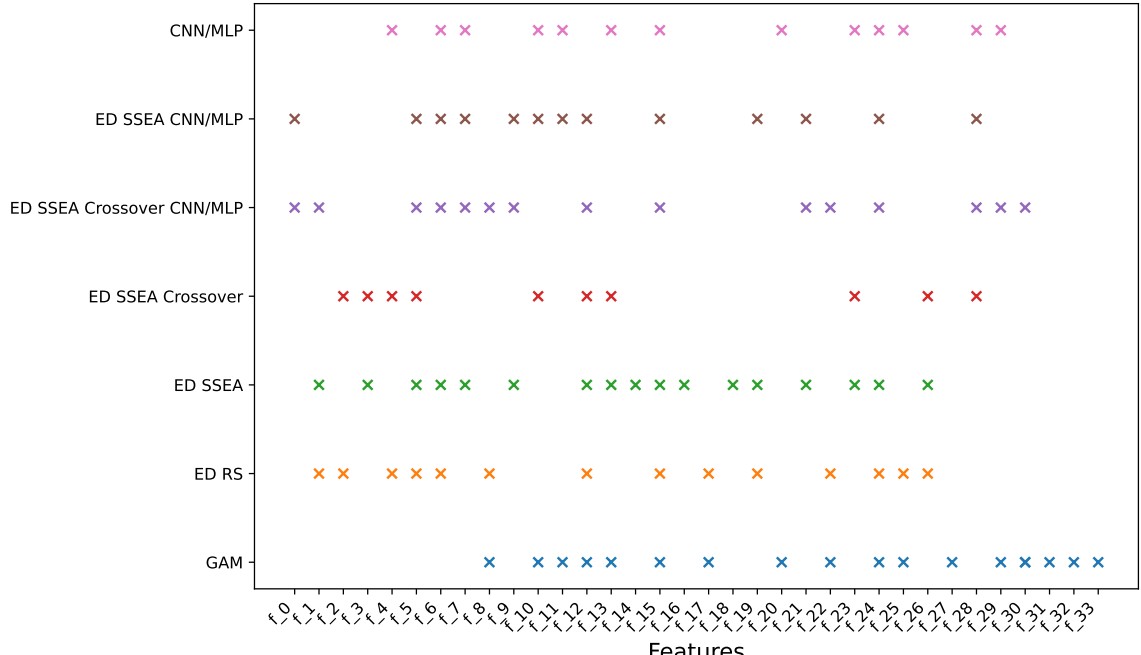

Figure 16: Features selected by the GAM, CNN/MLP and the various versions of EnergyDragon used in our experiments. The features name cannot be revealed due to the industrial confidentiality, and are renamed to $f_0, \ldots, f_{33}$.

## E  Norwegian Use Case

In this Section, we present the results of a reduced baseline on another case study: the hourly Norwegian Load for the year 2018.

### E.1  Dataset

The dataset comes from the ENTSO-E (European Network of Transmission System Operators for Electricity) transparency platform[5] and contains the Norwegian national load data at hourly intervals. Each day contains $H = 24$ time steps. We trained our models from 2014 to 2017 and compare the performance to the year 2018 using the MAPE ($\ell_{MAPE}$). For this dataset, we have access to 34 variables, including weather and calendar features. We have also anonymized these features in the following plots.

### E.2  Baseline

We reduced the full baseline presented in Section 4 by taking AutoPytorch with the traditional baseline (AutoPytorch), an open-source regression model, the Generalized Additive Model (GAM), and EnergyDragon with the steady-state algorithm, with and without the crossover (ED SSEA and ED SSEA Crossover). The setup for each model from the baseline is the same as in Section 4.

### E.3  Results

The results can be found in Table 3. They are similar to those found for the French use case. AutoPytorch with the traditional baseline is the worst model, with a MAPE 41% higher than the reference MAPE given by the GAM model. EnergyDragon was able to improve the GAM forecast even without the inputs from the CNN/MLP model. As in the French case, the crossover helped the

---

[5]https://transparency.entsoe.eu/load-domain/r2/totalLoadR2/show?

| Model | MAPE | RMSE (in MW) |
|---|---|---|
| GAM | 2.430% | 474.0 |
| AutoPytorch | 3.429% | 660.7 |
| ED SSEA | 2.196% | 465.7 |
| ED SSEA Crossover | **2.019%** | **426.3** |

Table 3: MAPE and RMSE of the different models from our baseline on theNorwegian dataset. The reference model is the GAM and the best model is highlighted in bold.

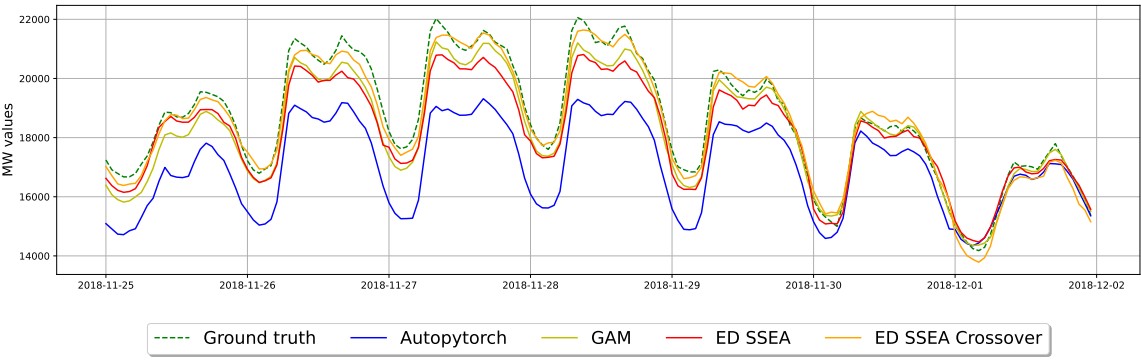

Figure 17: Norwegian load power forecasting for the last week of November 2018. The ground truth is displayed in dotted line.

algorithm to converge to a better result. This last model produced the best result, improving the GAM MAPE by 17%. The RMSE indicates that Norwegian residents consume less electricity than French residents. The comparison of the forecast for all models can be found Figure 17. All models got the correct curve shape, but except for the ED SSEA Crossover model, all models underestimated the load.

### E.4 EnergyDragon Results Analysis

Figure 18 and Figure 19 show the DNNs found by ED SSEA and ED SSEA Crossover respectively on the Norwegian dataset. The conclusions of Section D.1 do not hold anymore, since the best model found by ED SSEA Crossover does not use an attention layer. This dataset is a bit simpler, with only 24 values per day, resulting in a smoother signal than the French load with 48 values per day, and it can explain the use of Convolution layers instead of Attention layers to produce smoother predictions. The two DNNs are very different, with no common substructures, but compared to the french use case, the version with crossover did not produced an overly complicated DNN compared to the version without crossover. The features selected by both versions of EnergyDragon can be found Figure 20. The version without crossover selected 17 features, whereas the version with the crossover selected 13 features. Both models only have 6 features in common. Finally, Figure 21 shows the loss of the best model found so far over time for ED SSEA and ED SSEA Crossover. Both models converged in less than 5 hours, even if we let them run for another 20 hours. The crossover version converged very quickly, in less than an hour and a half. This fast convergence can explain the simple DNN compared to the DNNs obtained with the Crossover versions for the French use case. This good DNN was found before the algorithm used too many successive crossovers.

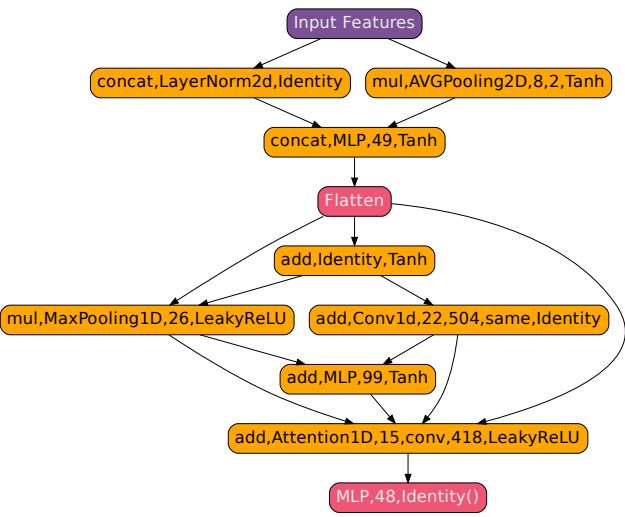

Figure 18: Architecture found by ED SSEA on the Norwegian dataset. Best MAPE=2.196%.

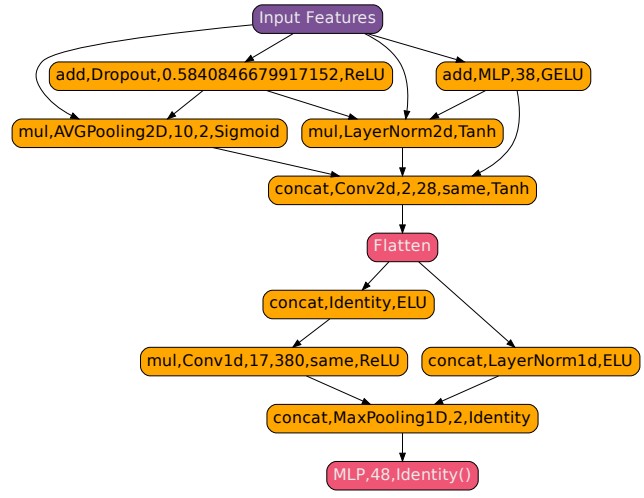

Figure 19: Architecture found by ED SSEA Crossover on the Norwegian dataset. Best MAPE=2.019%.

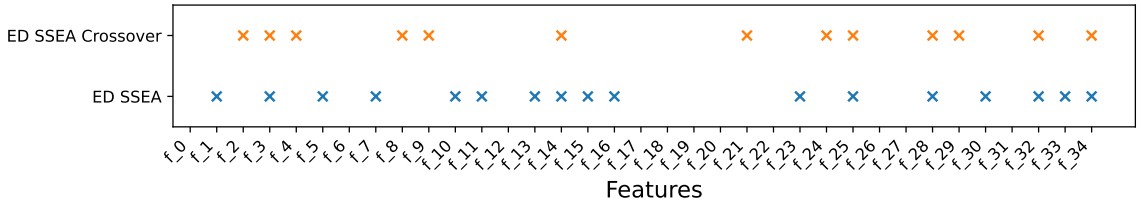

Figure 20: Features selected by ED SSEA and ED SSEA Crossover for the Norwegian dataset. The features name cannot be revealed due to the industrial confidentiality, and are renamed to. $f_0, \ldots, f_{34}$.

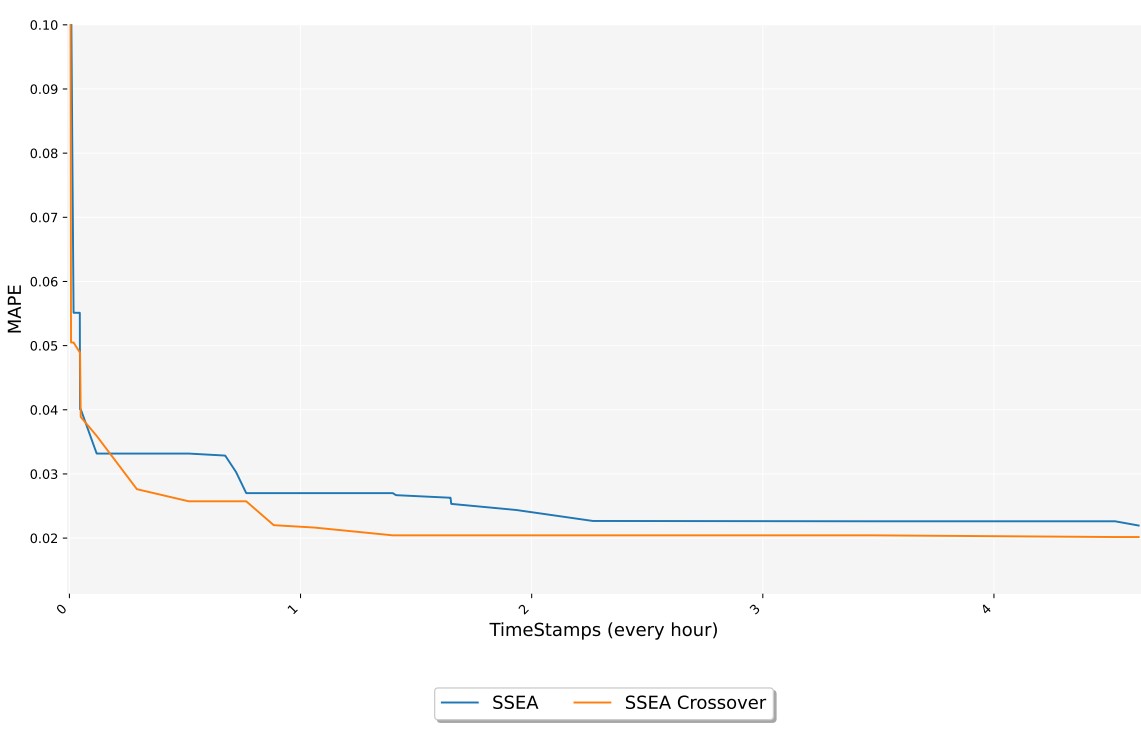

Figure 21: Loss of the best model over time for the different versions of EnergyDragon on the Norwegian dataset.

