# OpenReview forum: "Automated Deep Learning for load forecasting"
_automl.cc/AutoML/2024/ABCD_Track — AutoML 2024 (ABCD Track)_

### Official Review · Reviewer_f4Br · 2024-03-25

**Potential Impact On The Field Of Automl:** Good framework for a practical proble…
**Potential Impact On The Field Of Automl Rating:** 3
**Technical Quality And Correctness:** The paper is fairly technically sound…
**Technical Quality And Correctness Rating:** 3
**Clarity:** The paper is easy to read and overall…
**Clarity Rating:** 3
**Actions Required To Increase Overall Recommendation:** None

**Summary Of Contributions:**

The authors present a novel framework for AutoDL on electrical load forecasting, which is a time series forecasting problem with exogenous variables. they extend Dragon, to create their framework EnergyDragon.
The experimental results are convincing.

**Overall Review:**

Good paper.  simple but effective approach. The intuition that external features play an important load for electrical load forecasting is correct.

Some little things do need to be changed. Statements like these do not make sense:

1. ` The load signal can be explained almost entirely by a set of explanatory variables that do not include past data. Therefore state-of-the-art models tend to be based on regression techniques.`

"To improve the CNN/MLP model, we thought it might be interesting to use Automated Deep Learning"  The motivation is actually concerning. DL is powerful, but sometimes simpler models work well.The authors did not employ RandomForestRegressor or a XGBoostRegressor as a baseline. The GAM shows its effective, and given it is a rather less complex time series the reader wonders if the authors could have used off the shelf approaches. What about approaches such as ARIMAX ?

In Sec 3.32, Why use MSE? WMAPE or SMAPE can potentially be better  towards handling scale.

The experimental section is well presented, with a good set of baselines.

**Review Confidence:**

4

**Review Rating:**

8

**Review Summary:**

This is an appropriate paper for the track. There are some minor issues, which should be addressed.

---

### Official Review · Reviewer_bAQ3 · 2024-03-25

**Potential Impact On The Field Of Automl Rating:** 3
**Technical Quality And Correctness Rating:** 3
**Clarity Rating:** 3

**Summary Of Contributions:**

The paper introduces a novel AutoDL framework named EnergyDragon, aimed at enhancing electricity load forecasting. The authors address the importance of accurate load forecasting and the need to maintain grid stability, especially with the increasing integration of renewable energy sources. EnergyDragon selects features and optimizes the architecture and hyperparameters of DNNs, demonstrating superior performance over existing load forecasting methods, as well as AutoDL approaches on the French load signal dataset.

**Actions Required To Increase Overall Recommendation:**

- In practical energy management, updating forecasts in real-time based on incoming data is essential. Future work should address this dynamic scenario and the model's robustness to outliers, missing data, and ability to handle sudden changes in load patterns.
- The authors address the issue of interpretability with their approach. Are there any frameworks to address the importance of hyperparameters for this kind of framework to analyze the twenty-plus explanatory variables?

**Clarity:**

The paper's structure and writing style are suitable and easy to follow. The problem setting is explained sufficiently to follow the derivation of the EnergyDragon approach. The interpretation of the results is concise.

**Overall Review:**

Pros:
- The paper addresses a highly relevant and challenging problem in the energy sector.
- It introduces an innovative AutoDL framework that significantly outperforms existing models.
- The methodology is robust and features a novel feature selection method and an efficient search algorithm.
- Experimental results prove the viability of the approach against an RS baseline.

Cons:
- While the paper discusses the optimization of architecture and hyperparameters, a more detailed analysis of the hyperparameter tuning process could provide valuable insights since it is an integral part of AutoML.
- The experiments were not run on multiple seeds, which might impact the generalizability due to randomness in the data.

**Potential Impact On The Field Of Automl:**

EnergyDragon extends AutoDL's applicability to load forecasting. It showcases AutoDL's potential to tackle complex real-world problems. With their framework, the authors show that AutoML is usable for finding adaptive and efficient modeling strategies in the energy sector and other complex scenarios.

**Review Confidence:**

4

**Review Rating:**

8

**Review Summary:**

The paper introduces EnergyDragon, an AutoDL framework designed for electricity load forecasting.  Through feature selection and an asynchronous evolutionary search algorithm, EnergyDragon demonstrates a significant improvement over traditional regression-based models and other AutoDL approaches, as evidenced by experiments on the French load signal dataset. However, the paper could benefit from a more detailed exploration of the hyperparameter tuning process and the robustness of the model to dynamic real-world scenarios, such as real-time data updates and sudden changes in load patterns. Future work could also delve into the interpretability of models. Overall, the paper makes a valuable contribution to AutoDL for electricity load management and, thus, sustainability. I would therefore recommend to accept the paper.

**Technical Quality And Correctness:**

The authors give a clear explanation of the challenges in load forecasting and the motivation behind adopting AutoDL. The development of the EnergyDragon framework is well-articulated, from the definition of the search space to the feature selection method and the asynchronous evolutionary search algorithm. The experimental setup is thorough, including the dataset, baseline models, and comparative analysis.

---

### Official Review · Reviewer_Dd2a · 2024-03-26

**Potential Impact On The Field Of Automl Rating:** 4
**Technical Quality And Correctness Rating:** 4
**Clarity Rating:** 3

**Summary Of Contributions:**

The authors present EnergyDragon, a framework for performing NAS and HPO at the same time for energy load forecasting. Empirical evaluation on French energy load data shows the framework can achieve lower error values than other AutoML and state-of-the-art energy forecasting methods.

**Actions Required To Increase Overall Recommendation:**

Perform evaluations on a dataset from a different country and add open-source regression method baseline.

**Clarity:**

The paper is written well and clearly. However, some parts of the paper read more like a case study than a research paper, and a more formal writing style would be adviseable here (e.g. lines 52-53).

**Overall Review:**

Strengths:
- The paper deals with an interesting and important problem (energy forecasting) that has not yet been tackled to a great extent.
- The empirical evaluations clearly show that the framework outperforms all other methods for the given task, even state-of-the-art industry methods.

Weaknesses:
- Only the French load dataset is used for evaluating the framework. As different countries have different energy sources and usage, the results might not generalize to other countries.
- The regression-based models (GAM) mentioned in Table 1 are closed-source which hinders reproducibility. An evaluation on a comparable open-source variant (even if not state-of-the-art) would be helpful here.

**Potential Impact On The Field Of Automl:**

While most AutoML methods are tailored toward computer vision problems, the focus of this framework on tabular data from the energy forecasting domain provides a novel direction of research that should be explored in more detail in the future.

**Review Confidence:**

4

**Review Rating:**

8

**Review Summary:**

In summary, the framework presented in this paper tackles an interesting problem from an important domain while achieving good results. Generalizability could be improved further by performing evaluations on a different dataset.

**Technical Quality And Correctness:**

The proposed method implemented in the framework is technically correct and explained in great detail in the paper. The empirical results on real-world data show excellent performance.

---

### Meta-Review · Area_Chair_RBWK · 2024-04-22

**Paper Recommendation:** Accept
**Confidence:** 4

**Metareview:**

The paper introduces an AutoML framework for load forecasting.
The frameworks uses an evolutionary strategy to search for architectures and hyperparameters. Results are reported for a single seed on French load signal and also partly on Norwegian data in the appendix.

All reviewers recommended acceptance, noting the introduction of an interesting and important use-case at the intersection of Automl and time series forecasting.

Some reviewers also pointed a potential lack of baselines from simple statistics methods. After checking the paper, the experiments of the paper are indeed an area of concern:
* Clearly, some statistical baselines should be included, for instance a naive seasonal forecaster (which predicts the same hourly value as last week).
* Automl methods from time series method are entirely omitted whereas methods such as AutoGluon time series is likely to perform well in the studied setup and was shown to outperform a wide range of methods (https://arxiv.org/pdf/2308.05566.pdf)
* The evaluations are lacking in term of empirical evidence: a single seed is used whereas the performance of model is highly stochastic due to DL training and only two datasets are used. It is quite possible that the rank of methods would vary if other seeds, datasets, forecasting dates were used.

Given the unanimous reviews, I recommend acceptance also accounting for the impact of the application. However, I believe the paper impact could be improved by adding in the set of baselines AutoGluon as well as a simple baseline (seasonal naive or ensemble of statistical methods from statsforecast package).

---

### Decision · Program_Chairs · 2024-04-29

**Decision:**

Accept

**Comment:**

Thank you for submitting your paper. We are happy to tell you that we accept your paper to the main track. See you in Paris.